# Health Technology Reassessment: Addressing Uncertainty in Economic Evaluations of Oncology Drugs at Time of Reimbursement Using Long-Term Clinical Trial Data

Graeme Ball [1,*], Mitchell A. H. Levine [1,2], Lehana Thabane [1,2] and Jean-Eric Tarride [1,2,3]

[1] Department of Health Research Methods, Evidence, and Impact, Faculty of Health Sciences, McMaster University, Hamilton, ON L8S 4L8, Canada; levinem@mcmaster.ca (M.A.H.L.); thabanl@mcmaster.ca (L.T.); tarride@mcmaster.ca (J.-E.T.)

[2] The Research Institute of St. Joe's Hamilton, St. Joseph's Healthcare Hamilton, Hamilton, ON L8N 4A6, Canada

[3] McMaster Chair in Health Technology Management, McMaster University, Hamilton, ON L8S 4L8, Canada

* Correspondence: ballga@mcmaster.ca

**Abstract:** The evidence base to support reimbursement decision making for oncology drugs is often based on short-term follow-up trial data, and attempts to address this uncertainty are not typically undertaken once a reimbursement decision is made. To address this gap, we sought to conduct a reassessment of an oncology drug (pembrolizumab) for patients with advanced melanoma which was approved based on interim data with a median 7.9 months of follow-up and for which long-term data have since been published. We developed a three-health-state partitioned survival model based on the phase 3 KEYNOTE-006 clinical trial data using patient-level data reconstruction techniques based on an interim analysis. We used a standard survival analysis and parametric curve fitting techniques to extrapolate beyond the trial follow-up time, and the model structure and inputs were derived from the literature. Five-year long-term follow-up data from the trial were then used to re-evaluate the cost-effectiveness of pembrolizumab versus ipilimumab for treatment of advanced melanoma. The best fitting parametric curves and corresponding survival extrapolations for reconstructed interim data and long-term data reconstructed from KEYNOTE-006 were different. An analysis of the 5 year long-term follow-up data generated a base case incremental cost-effectiveness ratio (ICER) that was 28% higher than the ICER based on interim trial data. Our findings suggest that there may be a trade-off between certainty and the ICER. Conducting health technology re-assessments of certain oncology products on the basis of longer-term data availability, especially for those health technology adoption decisions made based on immature clinical data, may be of value to decision makers.

**Keywords:** oncology; reassessment; cost-effectiveness; survival analysis; uncertainty

## 1. Introduction

Drug reimbursement decision making often employs health technology assessments (HTAs) to detail the comparative value for money of one treatment versus another. HTA appraisals and funding recommendations are typically made on the basis of evidence from a single point in time when treatments enter the healthcare system. To enhance early access to novel health technologies, reimbursement decisions are increasingly made when the evidence base to support these decisions is lacking or far from mature [1], and HTA recommendations based on immature data or extrapolated short-term data often include the suggestion to collect additional data [2,3]. In recent years, HTA agencies have been advocating for a lifecycle management approach to health technology adoption and reimbursement decisions. In the United Kingdom, the National Institute for Health and Care Excellence (NICE) recently launched a 5 year strategy to adapt to a rapidly changing health and care landscape, which involves a more dynamic approach to health technology management [4]. The Canadian Agency for Drugs and Technologies in Health (CADTH)

has also been messaging the purported benefits of health technology management in which longer-term trial data and real-world evidence (RWE) could be used to re-assess already reimbursed drugs to ensure continued clinical and/or economic benefits are continuing to be realized by patients and in the marketplace. However, reimbursement decisions are rarely reconsidered, even once additional data have been collected. Publication of long-term follow-up data from clinical trials provides the opportunity to reassess decision making under substantially reduced clinical and economic uncertainty.

We present an economic evaluation of pembrolizumab for treatment of patients with advanced melanoma following ipilimumab therapy, which was studied in the KEYNOTE-006 phase 3 randomized controlled trial [5–7]. Pembrolizumab was approved by the Food and Drug Administration (FDA) [8] and recommended for reimbursement by HTA agencies including the NICE [9] and CADTH [10] using data from the first interim analysis (median duration of follow-up: 7.9 months) [5]. The CADTH recommendation [10] noted uncertainty in the modeling of long-term survival, stating "the original ipilimumab data demonstrated a sustained separation in the tail of the survival curve, a benefit that is yet to be confirmed in the pembrolizumab study". Similarly, the appraisal from NICE [9] noted that "the long-term benefits of pembrolizumab are highly uncertain". The evidence base for the approvals and reimbursement recommendations was based on short-term follow-up data with noted uncertainty, yet attempts to address this uncertainty do not seem to have been undertaken once the initial reimbursement decision was made. Despite the fact that an increasing number of HTAs published by national agencies are based on evidence that is assessed to be "uncertain", there is a paucity of available evidence for addressing this uncertainty in the context of reimbursement decision making.

Following publication of the FDA approval and several HTA agency recommendations [9,10], 5 year results from KEYNOTE-006 (median duration of follow-up: 57.7 months) [7] were published in a post hoc analysis of long-term follow-up data. The availability of this long-term follow-up data provided the opportunity to investigate the degree to which the results from a cost-effectiveness analysis based on interim data, where uncertainty is high, can accurately predict cost-effectiveness results based on longer-term data, where uncertainty is substantially reduced. To address this gap in the existing literature, we sought to determine the impact of using trial data of different maturity (long term versus short term) on survival curve extrapolations, and the impact of these different data on the results of a cost-effectiveness analysis, using the example of a pembrolizumab, which was reimbursed in several jurisdictions based on interim data.

## 2. Materials and Methods

A partitioned survival model was used to assess the cost-effectiveness of pembrolizumab versus ipilimumab for treatment of advanced melanoma from a US payer perspective over a 20 year time horizon. The results of 20 year survival extrapolations and cost-effectiveness based on interim data (median 7.9 months follow-up) were compared with the cost-effectiveness model based on the long-term follow-up data (median 57.7 months follow-up). Model inputs are presented in Table S1 in the Supplementary Materials.

### 2.1. Modeling Approach

We developed a three-health-state partitioned survival model (Figure 1) in Microsoft Excel® populated with two sets of data based on the published Kaplan–Meier (KM) curves for progression-free survival (PFS) and overall survival (OS) from the KEYNOTE-006 trial [5,7].

Progression-free survival (Figure 2) and OS (Figure 3) were extrapolated beyond the follow-up period of the trial using standard parametric curve fitting methods over a 20 year time horizon (a description of the extrapolation procedure is provided in Section 2.7).

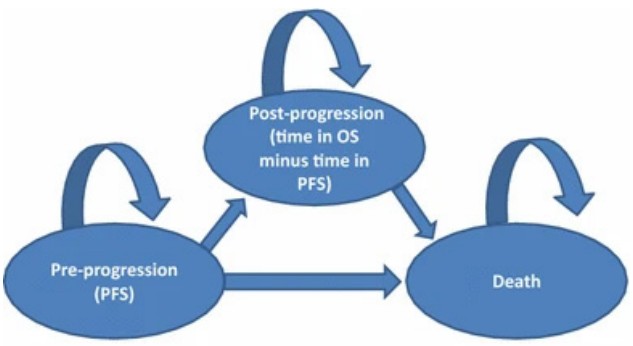

Abbreviations: PFS, progression-free survival; OS, overall survival.

**Figure 1.** Model structure and health states.

(**A**)

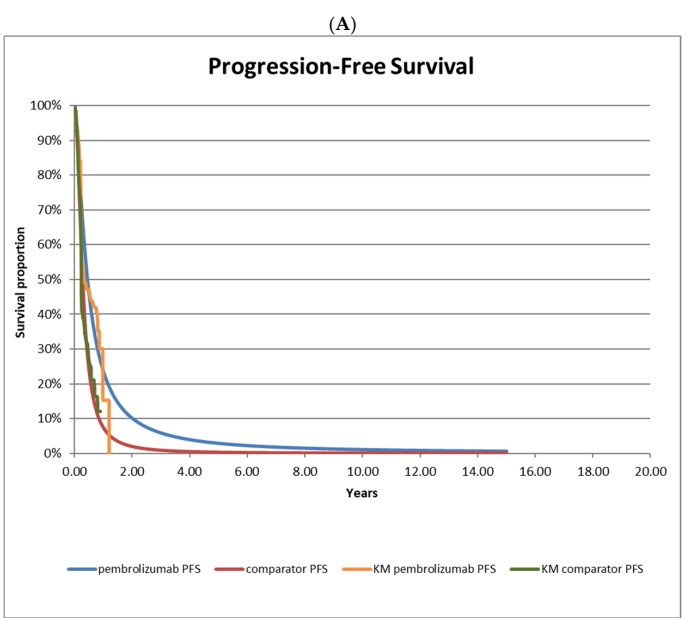

Abbreviation: KM, Kaplan–Meier; PFS, progression-free survival

(**B**)

Abbreviation: KM, Kaplan–Meier; PFS, progression-free survival

**Figure 2.** Survival curve extrapolations based on interim data from KEYNOTE-006. (**A**)—Progression-free survival based on interim data from KEYNOTE-006; (**B**)—Progression-free survival based on long-term follow-up data from KEYNOTE-006.

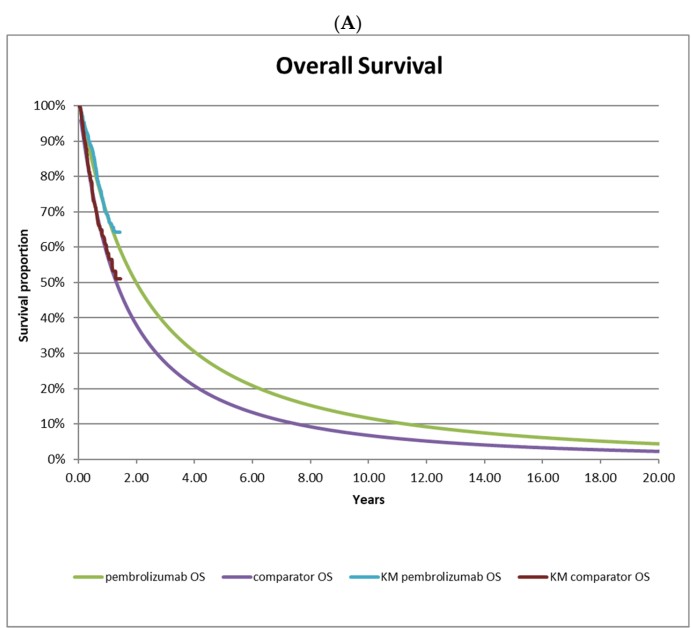

Abbreviations: KM, Kaplan–Meier; OS, overall survival; PFS, progression-free survival

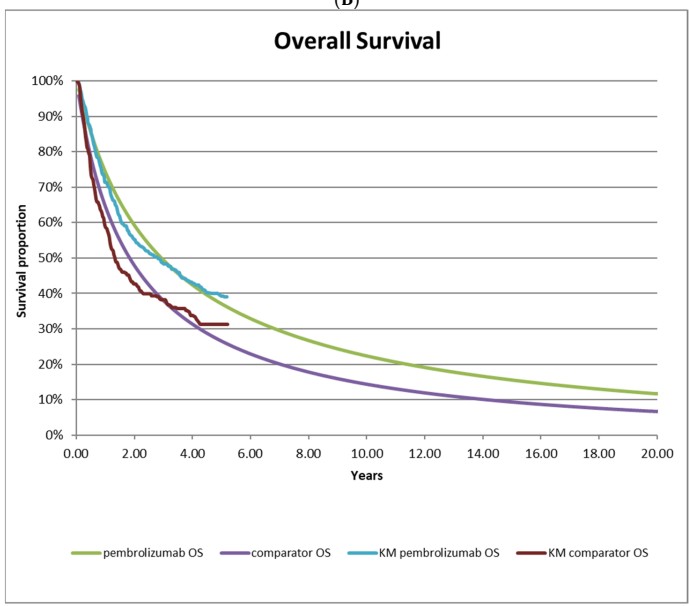

Abbreviations: KM, Kaplan–Meier; OS, overall survival; PFS, progression-free survival

**Figure 3.** Survival curve extrapolations based on long-term follow-up data from KEYNOTE-006. (**A**)—Overall survival based on interim data from KEYNOTE-006; (**B**)—Overall survival based on long-term follow-up data from KEYNOTE-006.

The starting age of the cohort was 63 years [5], and the cycle length was set as monthly. All patients entered the model through the pre-progression health state and could stay in this health state or transition either to the post-progression health state or to death according to transition probabilities calculated from reconstructed KM curves from KEYNOTE-006 [5]. Survival data for PFS and OS were used to determine the distribution of patients in the 'pre-progression' health state over time and the proportion of patients that transition to the 'death' health state for each treatment arm, respectively. The difference between the OS curve and the PFS curve yielded the proportion of patients experiencing progressive disease. The external validity of the modeling approach and survival analysis results were assessed through comparisons with pooled long-term ipilimumab data from patients with advanced melanoma reported by Schadendorf and colleagues [11] and through comparisons with a previously published cost-effectiveness analysis by Wang and colleagues [12] based on the interim data.

### 2.2. Clinical Inputs

The population modeled in our analyses included adult patients with advanced melanoma who were treated with either pembrolizumab or ipilimumab as depicted in the open-label, multicenter, randomized, controlled phase 3 KEYNOTE-006 trial. KEYNOTE-006 enrolled 834 patients, 556 of which were randomized to pembrolizumab and 278 to ipilimumab. The efficacy was analyzed in the trial according to the intention-to-treat population with OS and PFS as co-primary endpoints.

Adverse event rates of grade 3 or higher were modeled and sourced from the published clinical trial results based on interim data [5] and long-term follow-up data [7] from each of the KEYNOTE-006 treatment arms.

### 2.3. Regimen and Dosing

Dosing for ipilimumab was 3 mg/kg every 3 weeks up to a maximum of 4 doses, as per the FDA label. For pembrolizumab, the FDA-approved dosing of 2 mg/kg every 3 weeks was implemented in the model for a maximum of 2 years. In accordance with the KEYNOTE-006 trial protocol, a second course of pembrolizumab of up to 12 months was modeled for the proportion of patients who had not progressed by the end of 24 months of pembrolizumab treatment. The dose intensity was assumed to be 100%, and vial sharing was allowed in the base case (medication wastage was not explicitly accounted for).

### 2.4. Utility Values

Utility values were applied to each health state based on the EuroQoL five-dimension (EQ-5D) preference instrument values collected in KEYNOTE-006 and reported by Wang and colleagues [12]. Disutility adjustments were not made for adverse events, as these events were considered transitory and not anticipated to impact model results.

### 2.5. Healthcare Resource Utilization

Estimates of healthcare resource utilization associated with patient management in the pre-progression and post-progression health states were derived from the results of a US chart review study [13]. These estimates included oncologist visits, laboratory tests, and scans. Hospitalization costs for management of adverse events (grade 3 or higher) were estimated based on the proportions of patients experiencing grade 3 or higher adverse events reported in KEYNOTE-006 using Drug-Related Group (DRG) codes for gastrointestinal disorders, metabolism and nutrition disorders, and general disorders and administration site condition from the Centers for Medicare and Medicaid Services (CMS) final rule tables [14].

### 2.6. Costs

Unit costs for pembrolizumab and ipilimumab were based on the average sale price indicated in the 2023 Payment Allowance Limits for Medicare Part B Drugs sourced from CMS [15]. The average patient body weight used to model drug costs was back-calculated from the average dose of each drug reported in a previous cost-effectiveness analysis (98.7 kg for patients receiving ipilimumab, 112.0 kg for patients receiving pembrolizumab) [12]. Drug administration costs (per infusion) were derived from the CMS costs for hospital outpatient services list, and each drug infusion was assumed to incur a single administration cost [16]. The costs associated with patient management in the pre-progression and post-progression health states were estimated based on the results of a US chart review study, as were the costs associated with end of life [13]. The costs for subsequent therapies administered after progression were assumed to be the best supportive care in order to focus on comparative assessments between ipilimumab and pembrolizumab exclusively. No additional drug costs were modeled post-progression.

All costs were reported in USD 2023, and where necessary, costs derived from previous studies were inflated to USD 2023 using the US consumer price index [17].

*2.7. Statistical Analyses*

To populate the model, transition probabilities were estimated based on KM curves from KEYNOTE-006 which were digitized using Webplotdigitizer software (Version 4.6), and individual patient-level data were reconstructed according to the Guyot algorithm [18] using the statistical package R Studio. Standard parametric distributions (exponential, log-normal, log-logistic, gamma, Weibull, and Gompertz) were fitted to the reconstructed patient-level data and the statistical fit was assessed based on maximum likelihood estimation. Curve selection was based on the Akaike information criterion (AIC) as well as a visual inspection of the curves to assess the face validity of the fit (Table 1).

**Table 1.** Akaike information criterion values for parametric curve fitting.

| A—interim analysis data | | | | | | |
|---|---|---|---|---|---|---|
| **Parametric Curve Fits** | **Weibull** | **Exponential** | **Log-Normal** | **Log-Logistic** | **Gamma** | **Gompertz** |
| Ipilimumab OS-AIC | 933.1 | 932.7 | 921.7 | 927.6 | 932.0 | 934.6 |
| Ipilimumab PFS-AIC | 969.0 | 989.9 | 949.3 | 940.6 | 960.0 | 988.8 |
| Pembrolizumab OS-AIC | 818.4 | 821.0 | 813.6 | 816.1 | 817.6 | 822.2 |
| Pembrolizumab PFS-AIC | 964.9 | 965.1 | 947.8 | 947.4 | 962.4 | 965.6 |
| B—long-term follow-up data | | | | | | |
| **Parametric Curve Fits** | **Weibull** | **Exponential** | **Log-Normal** | **Log-logistic** | **Gamma** | **Gompertz** |
| Ipilimumab OS-AIC | 1583.2 | 1603.6 | 1542.8 | 1554.2 | 1591.6 | 1542.1 |
| Ipilimumab PFS-AIC | 1460.1 | 1471.7 | 1374.6 | 1370.4 | 1471.0 | 1404.0 |
| Pembrolizumab OS-AIC | 3189.3 | 3205.0 | 3140.5 | 3158.8 | 3196.2 | 3150.1 |
| Pembrolizumab PFS-AIC | 3277.5 | 3355.7 | 3178.0 | 3202.0 | 3304.0 | 3193.9 |

Abbreviations: AIC, Akaike information criterion; OS, overall survival; PFS, progression-free survival.

Validation of the extrapolated survival curves was undertaken through comparing estimated the life expectancy, hazard ratios, and the number of clinical events with the KEYNOTE-006 data (Table 2).

**Table 2.** Comparison of reconstructed Kaplan–Meier data versus KEYNOTE-006 data.

| A—interim analysis data | | | | | |
|---|---|---|---|---|---|
| **Data Source** | **Ipilimumab** | | **Pembrolizumab** | | **HR (95% CI)** |
| | **Median Survival** | **Events (N)** | **Median Survival** | **Events (N)** | |
| KEYNOTE-006 trial (OS) 2nd interim analysis | not reached | NR | not reached | NR | 0.69 (0.52–0.90) |
| Reconstructed (OS) | 15.7 | 112 | 24.0 | 90 | 0.67 (0.50–0.88) |
| KEYNOTE-006 trial (PFS) 2nd interim analysis | 2.8 | NR | 4.1 | NR | 0.58 (0.47–0.72) |
| Reconstructed (PFS) | 3.3 | 190 | 5.3 | 154 | 0.58 (0.47–0.72) |
| B—long-term follow-up data | | | | | |
| **Data Source** | **Ipilimumab** | | **Pembrolizumab** | | **HR (95% CI)** |
| | **Median Survival** | **Events (N)** | **Median Survival** | **Events (N)** | |
| KEYNOTE-006 trial (OS) long-term follow-up | 15.9 | 172 | 32.7 | 324 | 0.73 (0.61–0.88) |
| Reconstructed (OS) | 22.1 | 171 | 35.2 | 171 | 0.73 (0.59–0.90) |
| KEYNOTE-006 trial (PFS) long-term follow-up | 3.4 | 217 | 8.4 | 411 | 0.57 (0.48–0.67) |
| Reconstructed (PFS) | 4.8 | 222 | 10.9 | 402 | 0.55 (0.47–0.65) |

Abbreviations: CI, confidence interval; HR, hazard ratio; N, number; OS, overall survival; PFS, progression-free survival; NR, not reported.

For the cost-effectiveness analysis, the primary outcome of the analysis was calculated as the incremental cost per quality-adjusted life-year (QALY) gained, and both costs and outcomes were discounted at 3% annually as recommended by the Institute for Clinical and Economic Review (I.C.E.R.) [19]. Probabilistic sensitivity analyses were conducted using 1000 Monte Carlo simulations to account for parameter uncertainty. Cost-effectiveness acceptability curves [20] were generated to assess the probability of being cost-effective at varying willingness-to-pay (WTP) thresholds. Scenario analyses were also conducted to address structural uncertainty in the model. These scenario analyses included using the best fitting parametric functions identified in a previous cost-effectiveness analysis, a scenario in which all PFS and OS curves across both treatment arms were fitted with the Gompertz distribution, as was done in the pembrolizumab reimbursement submission to NICE. In addition, to address the difference in the average patient weight between the ipilimumab and pembrolizumab trial arms, an additional scenario analysis was run using the equal average patient weight across both treatment arms. Finally, we varied the proportion of patients remaining progression free who would receive pembrolizumab re-challenge after 2 years according to values cited in HTA agency recommendations.

## 3. Results

### 3.1. Survival Analysis of Interim Data [5] versus Long-Term Follow-Up Data [7]

Compared to the KEYNOTE-006 data, the number of events, hazard ratios, and median OS for pembrolizumab from the reconstructed data were found to closely replicate the trial results. However, the reconstructed data of the interim data overestimated the median PFS for both treatment arms and the median OS for ipilimumab (Table 2A). For the 5 year data, hazard ratios and the number of clinical events were consistent with the trial data. However, median survival outcomes were over-estimated in the long-term reconstructed data (Table 2B).

To extrapolate the PFS and OS data over the 20 year time horizon using the interim data, the best-fitting curves for pembrolizumab and ipilimumab PFS and OS were the log-logistic and log normal distributions for both treatment groups, respectively, and log-normal for both treatment arms in the interim data (Table 2A). For KEYNOTE-006 long-term follow-up data, the best-fitting parametric survival curves were found to be log-logistic for pembrolizumab PFS, log-normal for pembrolizumab OS, Gompertz for ipilimumab PFS, and log-normal for ipilimumab OS (Table 2B).

### 3.2. Cost-Effectiveness Analysis of Interim Data [5] versus Long-Term Follow-Up Data [7]

Using the interim data, pembrolizumab generated a total of 3.99 undiscounted life-years (LYs) compared to 2.85 undiscounted LYs for ipilimumab. The probabilistic ICUR was USD 100,293 per QALY gained (deterministic ICUR: USD 111,861). Pembrolizumab was found to be cost-effective in 97% of the simulations at a commonly cited WTP threshold of USD 150,000/QALY gained and 51% of simulations at a WTP threshold of USD 100,000/QALY gained (Figure 4A). The model was most sensitive to cost-related inputs, including the discount factor, cost of pembrolizumab, and average patient weight (Figure S1 in the Supplemental Materials). The results of the scenario analyses are presented in Table S2 in the Supplemental Materials.

Incorporating the reconstructed long-term follow-up data into the model generated 5.91 undiscounted LYs for pembrolizumab, and 4.31 undiscounted LYs for ipilimumab. The probabilistic ICUR was estimated to be USD 139,583 per QALY gained (deterministic ICUR: USD 156,829/QALY gained). At a WTP of USD 150,000/QALY gained, pembrolizumab was cost-effective in 66% of the simulations and was cost-effective in 3% of simulations at a WTP threshold of USD 100,000/QALY gained (Figure 4B). The ICUR was found to be sensitive to the clinical inputs, including the shape parameter for both pembrolizumab and ipilimumab, but was also sensitive to the discount factor (Figure S1 in the Supplemental Materials). The scenario analysis results are presented in Table S2 in the Supplemental Materials.

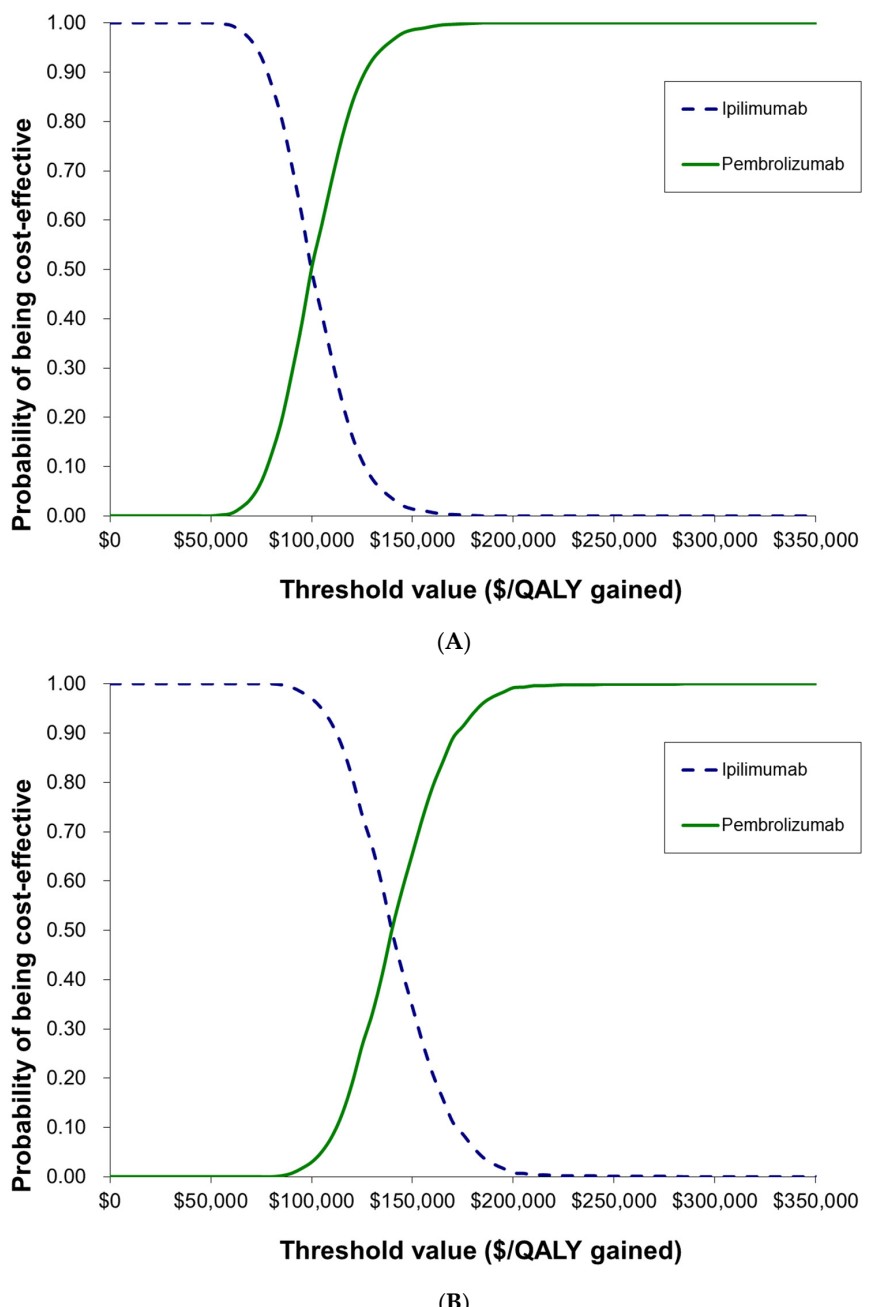

**Figure 4.** Cost-effectiveness acceptability curves. (**A**)—cost-effectiveness acceptability curves based on interim data from KEYNOTE-006; (**B**)—cost-effectiveness acceptability curves based on long-term follow-up data from KEYNOTE-006.

## 4. Discussion

To our knowledge, this is the first reassessment of a drug reimbursed based on interim data and for which long-term data have been published thereafter. Here, we conducted an economic evaluation of pembrolizumab versus ipilimumab for advanced melanoma using two reconstructed datasets generated from the KEYNOTE-006 trial interim analysis and a post hoc long-term follow-up analysis. We used commonly reported modeling approaches and standard parametric curve fitting techniques to extrapolate the clinical trial results over a 20 year time horizon and to estimate the cost-effectiveness of pembrolizumab versus ipilimumab from a US payer perspective. Our re-analysis using the long-term follow-up data generated an ICUR that was 42% higher than the ICUR based on the interim trial data. There are a number of reasons that may explain this result. First, the shapes of the KM

survival curves are only partially known with interim data, and the shape of the survival curves used for long-term extrapolated outcomes may change as more data are collected, as shown in KEYNOTE-006. This additional data could explain why different best-fitting parametric curves were observed in our analysis based on long-term data compared with the analysis based on interim data and why the long-term survival extrapolations vary between our two models.

As clinicians gain experience with new drug indications, such as pembrolizumab for advanced melanoma, clinical management is likely to incrementally improve, and such improvements might not be fully reflected in KM survival curves based on interim trial data. In KEYNOTE-006, a notable shift outward in the KM PFS curves was reported in the long-term follow-up data (mPFS 8.4 months) compared with the published interim data (mPFS 4.1 months) (Figure 2). This outward shift also impacted our scenario analyses for pembrolizumab as treat-to-progression, showing a much larger percent change from the base case in the analysis based on long-term follow-up data (Table S2 in the Supplementary Materials). While the exact reasons for this outward shift may not be clear, improvements in clinical management could be an explanatory factor. The long-term OS data for ipilimumab presented in KEYNOTE-006 (31% patients alive at 5 years) were also notably higher than the OS data (22% patients alive at 5 years) from a long-term pooled analysis [11] based on 10 previous ipilimumab phase 2 and phase 3 trials (Figure S2 in the Supplementary Materials). Data from the pooled analysis were based on trials published between 2010 and 2013, whereas the interim data from KEYNOTE-006 were published in 2015. The outward-shifted PFS curves presented in the long-term KEYNOTE-006 data could be at least partially explained by improvements in the clinical management of advanced melanoma patients.

The differences we observed in our interim data- and long-term data-based cost-effectiveness analyses suggest a tradeoff between the need to make recommendations for patient access based on uncertain clinical benefits from shorter term (immature) data, and the delay that would be required to make decisions based on longer-term clinical data in which uncertainty is substantially reduced. While coverage with evidence development has been considered by some HTA bodies as a means of addressing this inherent tradeoff, the availability of long-term data may be sufficiently impactful to warrant an HTA reassessment.

### 4.1. Previous Studies

A previous cost-effectiveness analysis [12] used the KM curves from the trial for the first 60 weeks, then used parametric extrapolations based on a previous study by Schadendorf and colleagues [11] from 20 to 260 weeks for the ipilimumab arm. For the pembrolizumab arm, the authors used trial data for the first 60 days, then applied a time-varying hazard ratio versus ipilimumab between week 60 and 260 based on a previous study [11], and then used data from a US melanoma registry by Balch and colleagues [21] thereafter. In the pembrolizumab recommendation from NICE for treatment of advanced melanoma, the Evidence Review Group stated that there was a risk of selection bias in using data from the Schadendorf study for extrapolation, as well as limitations in the algorithm used to adjust for patient characteristics and the long-term survival data from Balch and colleagues [21] used to project long-term survival. In contrast, we used trial data only to perform survival curve extrapolations, a simplified approach which nevertheless closely replicated the incremental LY estimates reported in the previous cost-effectiveness study [12] over a 20 year time horizon. Our estimate of an undiscounted incremental gain of 1.15 LYs for treatment of pembrolizumab over ipilimumab was very close to the number reported in a previous cost-effectiveness study (1.14 LYs). These findings suggest that our survival curve extrapolation approach may be appropriate.

However, our base case probabilistic ICUR estimate based on interim data from KEYNOTE-006 (USD 111,861/QALY gained) was higher than in the previous CEA based on the same interim data (USD 81,091/QALY gained). The most likely explanation for this discrepancy may be found in the differences in how utility values were applied. We applied

EQ-5D utility values to the progression-free (0.83) and progressed (0.78) health states, whereas the authors of the previous analysis calculated utility scores based on multiple time-to-death categories: 360 days or more (0.85), 270–360 days (0.78), 180–270 days (0.74), 90–180 days (0.75), 30–90 days (0.69), and under 30 days (0.48) to death. Given that our estimates of incremental undiscounted LYs (1.15 vs. 1.14) and incremental discounted total costs (USD 59,023 vs. USD 63,680) were nearly identical to those of Wang et al. (2015), we conclude that the differences in our results can be largely accounted for by the differences in how utility values were applied in the models.

### 4.2. Strengths

A number of strengths can be identified in our approach and results. First, the modeling approach and non-clinical input parameters were identical for both the analyses based on interim data and analyses based on long-term data, allowing us to isolate the impact of survival data on the model results. Second, we utilized the same partitioned survival modeling approach reported in HTA agency appraisals that have reviewed KEYNOTE-006 data for advanced melanoma, which helps to support the external validity of our results. We also derived relevant cost inputs from published real-world data [13] and up-to-date CMS costing databases [14–16], as well as a previously published economic evaluation [12] in order to enhance external validity and comparability with previously conducted studies and HTAs. These inputs and methods allowed us to produce an updated estimate of the cost-effectiveness of pembrolizumab versus ipilimumab for advanced melanoma based on more recent clinical and economic evidence than has been previously published.

Another advantage of our analysis concerns the use of long-term follow-up data in which a large number of patients at risk was retained throughout the vast majority of the long-term follow-up period. In general, it is more challenging to validate results based on small sample sizes, and our use of long-term follow-up data provided a sufficient sample size to have confidence in the results despite a high degree of censoring in the tail of the survival curves.

When modeling chronic conditions (such as cancer), or when treatments have differential effects on mortality, a lifetime horizon is most appropriate. Our survival extrapolations, based on published clinical data from the KEYNOTE-006 trial, indicate that approximately 10% of patients are expected to be alive at the 20 year timeframe (Figure 3B). Using a shorter time horizon would result in important clinical events (e.g., disease progression and death) being missed, and the full costs and clinical benefits would not be captured. While a time horizon of 20 years may seem long, it is consistent with the published literature [12] and health technology appraisal documents from NICE [9] and CADTH [10]. A shorter time horizon would therefore not be appropriate.

In addition, we based our extrapolation approach on the NICE DSU TECHNICAL SUPPORT DOCUMENT 21: Flexible Methods for Survival Analysis [22]. The long-term hazards are expected to follow a simple shape in KEYNOTE-006, exemplified in the simple shape (no kinks and no inflection points) of the long-term OS curve reported by Robert 2019 [7] (5 years follow-up), indicating that the standard parametric curve-fitting approach is appropriate. In addition, although other approaches could be used to model outcomes from KEYNOTE-006, our model follows methodological recommendations from the NICE DSU 21 and is aligned with models from the UK [9], Canada [10], and the US [12].

### 4.3. Limitations

Our study is not without limitations. First, we did not have access to the trial patient level data and we relied on reconstructing KM curves using digitization techniques. However, our reconstructed survival data closely replicated the hazard ratios and number of clinical events reported in the interim and long-term data from KEYNOTE-006. While the reconstructed data consistently over-estimated median survival outcomes, the survival over-estimation was consistent when using either the interim or long-term data (Table 2), and therefore it should not affect our primary conclusions. Second, several of our model

input parameters, including healthcare resource utilization and utility values, were derived from model parameters presented in work by Wang et al. [12]. However, we did not have access to their model, which could explain why the base case ICUR in our cost-effectiveness analysis based on interim data was higher than what has been reported in previous research [12]. In addition, we based our analyses on clinical trial data, and health technology reassessments using RWE could be used as an additional confirmatory source of evidence.

Another limitation of our approach is that during the time period between publication of interim trial data and the subsequent availability of long-term follow-up data, new comparators may have arisen in the clinical environment which may render the results of a re-assessment using the same comparison less clinically relevant. Finally, two different doses of pembrolizumab were studied in the interim analysis of KEYNOTE-006, whereas we modeled only the 2 mg/kg every 3 weeks dose in order to align with the FDA-approved dose. Outcomes from both doses were reported in a combined KM curve for PFS and OS in the long-term pembrolizumab data, but since the two doses studied in the interim analysis had overlapping PFS and OS curves, the impact of this discrepancy is expected to be minimal.

Although HTAs are not commonly used in the US for decision making compared to countries with well-established HTA systems such as the UK or Canada, we replicated a US cost-effectiveness model in the absence of published cost-effectiveness studies of pembrolizumab for treatment of patients with advanced melanoma in Canada and the UK. Using this US study as a foundation for model inputs, we replicated the US model as closely as possible and therefore we also used a US payer perspective. While the US publication provided a lot of information on the methods and model inputs (e.g., OS extrapolation), some details regarding some model parameters (e.g., utility data) were not available in the US publication, which explains why we were not able to replicate the exact results of the US study. Nonetheless, our primary aim was to replicate the survival reported in the previous study, which is a critical validation step before conducting our re-assessment using long-term data, and we achieved this aim very closely; the number of incremental LYs estimated in our study (1.15) was nearly identical to the value reported in the Wang 2017 study (1.14). Our incremental cost estimate (USD 59,023) also closely matched the previous study (USD 63,680), providing additional validation for our replication approach.

### 4.4. Future Research

We have demonstrated the impact of long-term clinical trial data on the results of a cost-effectiveness analysis for a single drug in a single therapeutic indication. While the interim analysis from KEYNOTE-006 provided promising preliminary data, the latter half of the KM curves were heavily censored, leading to clinical uncertainty and rendering validation difficult. We addressed this uncertainty by conducting a re-assessment based on long-term follow-up data. However, since the clinical environment of phase 3 trials is highly controlled, our trial-data-based results could be validated in future studies through incorporation of real-world clinical evidence.

Standard parametric curve fitting and extrapolation were used in our analyses. However, a plateau was observed in the long-term follow-up data from both arms of KEYNOTE-006, as well as from long-term pooled data from previous ipilimumab studies [11], implying a potentially curative effect for a small but defined proportion of the advanced melanoma patient population treated with either pembrolizumab or ipilimumab. To account for these observed survival plateaus in the long-term data, a mixture–cure modeling approach [23], in which the patient populations are stratified into 'cured' and 'non-cured' groups to better capture their respective outcomes, could be more methodologically appropriate than standard parametric curve fitting. Future research is encouraged to investigate the impact of using this approach. In addition, our reconstructed data, while very accurately replicating the hazard ratios and number of events reported in KEYNOTE-006, nevertheless overestimated the median PFS and the median OS for the ipilimumab treatment arm (Table 2). This may be at least in part due to the shape of the KM curves which had a

steep drop around month 3, resulting in an "s"-shaped curve that is challenging to fit with a single parametric distribution. As a result, fitting spline models with several knots to the reconstructed trial data could potentially be a reasonable alternative methodology to consider in future research.

## 5. Conclusions

While clinical and economic uncertainty may be reduced with longer-term follow-up data, the results of our analysis suggest that this reduction may come at a cost: decreased cost-effectiveness. Our findings suggest that there may be good reason to consider conducting health technology re-assessments of certain oncology products on the basis of longer-term data availability, especially for those health technology adoption decisions made based on immature clinical data. A lifecycle or health technology management approach could be a practical solution for decision makers to ensure that decision making remains informed by the most appropriate, relevant, and up-to-date evidence. Future research comparing cost-effectiveness models based on interim and final data would be required to generalize the results of our study to other settings.

**Supplementary Materials:** The following supporting information can be downloaded at: https://www.mdpi.com/article/10.3390/curroncol30070484/s1, Figure S1: Deterministic sensitivity analyses; Figure S2: Overall survival based on long-term follow-up data from KEYNOTE-006 and pooled long-term ipilimumab data from Schadendorf et al., 2015; Table S1: Model input parameter values; Table S2: Scenario analyses.

**Author Contributions:** Conceptualization, G.B. and J.-E.T.; methodology, G.B., M.A.H.L., L.T. and J.-E.T.; validation, G.B., M.A.H.L., L.T. and J.-E.T.; formal analysis, G.B.; investigation, G.B.; data curation, G.B.; writing—original draft preparation, G.B. and J.-E.T.; writing—review and editing, G.B., M.A.H.L., L.T. and J.-E.T.; supervision, J.-E.T. All authors have read and agreed to the published version of the manuscript.

**Funding:** This research received no external funding.

**Institutional Review Board Statement:** Not applicable.

**Informed Consent Statement:** Not applicable.

**Data Availability Statement:** The data presented in this study are available upon reasonable request from the corresponding author.

**Conflicts of Interest:** Graeme Ball is an employee of Gilead Sciences, Inc. (Canada). Jean-Eric Tarride, Mitch Levine, and Lehana Thabane have no conflict of interest that are directly relevant to the content of this article.

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
