# Peer review of "Health Technology Reassessment: Addressing Uncertainty in Economic Evaluations of Oncology Drugs at Time of Reimbursement Using Long-Term Clinical Trial Data"

_curroncol, doi:10.3390/curroncol30070484_

Round 1

Reviewer 1 Report (Previous Reviewer 2)

Thank you for giving me the opportunity to review the article (resubmitted manuscript of the curroncol-2351584). The authors corrected the manuscript according to the comments. Therefore, the reviewer thought that the manuscript can be accepted for publication in the journal.

None

Reviewer 2 Report (Previous Reviewer 1)

I agree with the revised version.

A single minor remark, Line 406: the term "HTA countries" is not broadly used, suggest replacing e.g., by "countries with well established health technology assessment systems". 

This manuscript is a resubmission of an earlier submission. The following is a list of the peer review reports and author responses from that submission.

Round 1

Reviewer 1 Report

This is a very important paper about possible health technology reassessment of oncology drugs that were initially reimbursed based on short-term / interim study results, when new longer-term clinical data becomes available. A case study is presented, reassessing the incremental cost/utility ratio of pembrolizumab for patients with advanced melanoma.  

Before publication, please consider revising the following parts:

- The Authors showed that NICE and CADTH recommended pembrolizumab for reimbursement in this indication based on clinical and health economic evidence, while no health economics focused assessment is cited from the US. Relying on health utility, QALY, or ICER/ICUR in reimbursement decisions is not well established in the US. Hence, it is unclear why is the reassessment focusing on the US payer perspective (Line 258). Instead, focusing on the reassessment of prior NICE and CADTH assessments would be more useful / easier to put in context. If the reassessment perspective remains to focus on the US, justification is recommended.   

- The Authors stated that their base case probabilistic ICUR estimate was higher than expected from a previous modelling study based on the same interim data, and argued that the difference was explained by different estimation of utility values.  (Lines 320-321). This is a key limitation in my view: during reassessment, only those model inputs are expected to be changed which could be replaced by more relevant / less uncertain data. It is unclear why the health utility calculation methods are diverging from a (the ) reference model. If not properly justified, this parallel methodology adjustment undermines the conclusion on higher ICUR value when using long-term clinical outcomes instead of interim / short-term clinical results.  

Author Response

Reviewer #1

Comments and Suggestions for Authors

This is a very important paper about possible health technology reassessment of oncology drugs that were initially reimbursed based on short-term / interim study results, when new longer-term clinical data becomes available. A case study is presented, reassessing the incremental cost/utility ratio of pembrolizumab for patients with advanced melanoma.  

Before publication, please consider revising the following parts:

Reviewer’s comment #1: The Authors showed that NICE and CADTH recommended pembrolizumab for reimbursement in this indication based on clinical and health economic evidence, while no health economics focused assessment is cited from the US. Relying on health utility, QALY, or ICER/ICUR in reimbursement decisions is not well established in the US. Hence, it is unclear why is the reassessment focusing on the US payer perspective (Line 258). Instead, focusing on the reassessment of prior NICE and CADTH assessments would be more useful / easier to put in context. If the reassessment perspective remains to focus on the US, justification is recommended.   

Authors’ answer to comment #1: We thank the reviewer for this important question. In the absence of Canadian or UK published cost-effectiveness model of pembrolizumab for the treatment of patients with advanced melanoma, our study was modeled on a published cost-effectiveness analysis from the US payer perspective (Wang 2017) based on the interim analysis from KEYNOTE-006. Using this as a foundation for model inputs, we replicated the US model as closely as possible and therefore we also use a US payer perspective.

In addition, global cost-effectiveness analyses typically use the US as the reference payer perspective and therefore US modeling studies can be expected to have a greater reach and audience interest than a specific country-level adaptation.

To address this comment and the next comment, we added the following text in lines 406-420 of the Discussion section.

“Although HTA is not commonly used in the US for decision making compared to HTA countries such as the UK or Canada, we replicated a US cost-effectiveness model in the absence of published cost-effectiveness studies of pembrolizumab for treatment of patients with advanced melanoma in Canada and the UK. Using this US study as a foundation for model inputs, we replicated the US model as closely as possible and therefore we also used a US payer perspective. While the US publication provided a lot of information on the methods and model inputs (e.g. OS extrapolation), some details regarding some model parameters (e.g. utility data) were not available in the US publication which explains why we were not able to replicate the exact results of the US study. Nonetheless, our primary aim was to replicate the survival reported in the previous study, which is a critical validation step before conducting our re-assessment using long-term data, and we achieved this aim very closely: the number of incremental LYs estimated in our study (1.15) was nearly identical to the value reported in the Wang 2017 study (1.14). Our incremental cost estimate ($59,023) also closely matched the previous study ($63,680), providing additional validation for our replication approach.”

Reviewer’s comment #2: The Authors stated that their base case probabilistic ICUR estimate was higher than expected from a previous modelling study based on the same interim data, and argued that the difference was explained by different estimation of utility values.  (Lines 320-321). This is a key limitation in my view: during reassessment, only those model inputs are expected to be changed which could be replaced by more relevant / less uncertain data. It is unclear why the health utility calculation methods are diverging from a (the ) reference model. If not properly justified, this parallel methodology adjustment undermines the conclusion on higher ICUR value when using long-term clinical outcomes instead of interim / short-term clinical results. 

Authors’ answer to comment #2: The reviewer raises a good point regarding utility values. While we based our model on a previously published cost-effectiveness analysis (Wang 2017), we were not able to replicate the complex approach for utility values used in the previous study because we did not have access to the model developed by the previous study’s authors.

As stated in our response to the reviewer’s previous comment, we added explanatory text to lines 406-420 in the Discussion section (please see paragraph above). No additional changes were made to the paper other than providing the rationale for using the US cost-effectiveness study as per the previous answer. 

Reviewer 2 Report

Thank you for giving me the opportunity to review the article. The author conducted a study focusing on the health technology reassessment. The topic is socially important, but there are fundamental methodological problems in the manuscript. Therefore, the reviewer thought that the manuscript cannot be accepted for publication. I listed my comments below.

Comments:

Abstract:

1.      This study only focusing on a drug, but the conclusion statement is too general.

Methods:

2.      The authors should justify to set the 20-year time-period. The detailed explanation is important.

3.      The validity of the 20-year survival extrapolations (standard parametric curve-fitting methods for this kind of diseases) is unclear.

4.      The scenario the authors used for the long-term evaluation is strange. For example, the mean age of the KEYNOTE-006 Study participants is 60.1 years, and 20 years is an inappropriate extrapolation period for such a population.

5.      The model which the authors used is too simple, and it is difficult to evaluate the long-term changes of patients’ status.

None.

Author Response

Reviewer #2

Comments and Suggestions for Authors

Thank you for giving me the opportunity to review the article. The author conducted a study focusing on the health technology reassessment. The topic is socially important, but there are fundamental methodological problems in the manuscript. Therefore, the reviewer thought that the manuscript cannot be accepted for publication. I listed my comments below.

 Comments:

Abstract:

  1. Reviewer’s comment #1: This study only focusing on a drug, but the conclusion statement is too general.

Authors’ response to comment #1: We thank the reviewer for this comment. While we believe that our conclusion logically follows from the results we presented, nevertheless we have modified our final paragraph (underlined text) in response to the reviewer’s comment to reflect a more precise conclusion:

“While clinical and economic uncertainty may be reduced with longer-term follow-up data, the results of our analysis suggest that this reduction may come at a cost: decreased cost-effectiveness. Our findings suggest that there may be good reason to consider conducting health technology re-assessments of certain oncology products drugs on the basis of longer-term data availability, especially for those health technology adoption decisions made based on immature clinical data. A lifecycle or health technology management approach could be a practical solution for decision makers to consider to ensure that oncology drug decision making remains informed by the most appropriate, relevant, and up-to-date evidence. Additional research comparing cost-effectiveness models based on interim and final data would be required to generalize the results of our study to other settings.

Methods:

  1. Reviewer’s comment #2: The authors should justify to set the 20-year time-period. The detailed explanation is important.

Authors’ response to comment #2: Thank you for this comment. We based our study on a previously published US cost-effectiveness analysis in which a 20-year time horizon was used (Wang 2017). In order to replicate the results of this previous study, we utilized a 20-year time horizon.

When modelling chronic conditions (such as cancer), or when treatments have differential effects on mortality, a lifetime horizon is most appropriate (guidelines from NICE, CADTH, and I.C.E.R.). Our survival extrapolations, based on published clinical data from the KEYNOTE-006 trial, indicate that approximately 10% of patients are expected to be alive at the 20-year timeframe. Using a shorter time horizon would result in important clinical events (e.g. disease progression, death) being missed, and the full costs and clinical benefits would not be captured. Our time horizon of 20 years is therefore a conservative approach, and a shorter time horizon would not be appropriate.

To fully reflect the above points in the manuscript, we have added the following text to lines 363-371 of the Discussion section:

“When modelling chronic conditions (such as cancer), or when treatments have differential effects on mortality, a lifetime horizon is most appropriate. Our survival extrapolations, based on published clinical data from the KEYNOTE-006 trial, indicate that approximately 10% of patients are expected to be alive at the 20-year timeframe (Figure 3B). Using a shorter time horizon would result in important clinical events (e.g. disease progression, death) being missed, and the full costs and clinical benefits would not be captured. While a time horizon of 20 years may seem long, it is consistent with published literature[12] and health technology appraisal documents from NICE[9] and CADTH[10]. A shorter time horizon would therefore not be appropriate.”

  1. Reviewer’s comment #3: The validity of the 20-year survival extrapolations (standard parametric curve-fitting methods for this kind of diseases) is unclear.

Authors’ response to comment #3: We appreciate the reviewer’s request for clarification regarding standard curve-fitting methods. In response, we have added the following text to lines 101-103 to direct the reader to where we present details of the curve-fitting procedure:

“Progression-free survival (Figure 2) and OS (Figure 3) was extrapolated beyond the follow-up period of the trial using standard parametric curve-fitting methods over a 20-year time horizon (description of extrapolation procedure provided in section 2.7).”

To provide additional clarity regarding our approach, we followed the NICE DSU TECHNICAL SUPPORT DOCUMENT 21: Flexible Methods for Survival Analysis which states that “[s]tandard parametric models can provide reasonable extrapolations if long-term hazards are expected to follow simple shapes”. As the long-term follow-up data from KEYNOTE-006 was reported in Robert 2019, we can observe the simple shape (no kinks, no inflection points) of the OS curve at 5 years of follow-up, indicating that the parametric curve-fitting approach is appropriate.

To address the validity of the 20-year time horizon in the manuscript, we added the following text to lines 372-380 in the Discussion section:

“In addition, we based our extrapolation approach on the NICE DSU TECHNICAL SUPPORT DOCUMENT 21: Flexible Methods for Survival Analysis[23]. The long-term hazards are expected to follow a simple shape in KEYNOTE-006, exemplified in the simple shape (no kinks, no inflection points) of the long-term OS curve reported in Robert 2019[7] (5 years follow-up), indicating that the standard parametric curve-fitting approach is appropriate. In addition, although other approaches could be used to model outcomes from KEYNOTE-006, our model follows methodological recommendations from the NICE DSU 21 and is aligned with models from the UK (NICE HTA), Canada (CADTH HTA), and the US (Wang 2017).”

Finally, since we developed our analysis based on the published cost-effectiveness analysis by Wang 2017, it was important to ensure that we followed the same modeling approach.

Based on the rationale in the paragraphs above, we believe the standard parametric curve-fitting approach is fully justified for our analysis.

  1. Reviewer’s comment #4: The scenario the authors used for the long-term evaluation is strange. For example, the mean age of the KEYNOTE-006 Study participants is 60.1 years, and 20 years is an inappropriate extrapolation period for such a population.

Authors’ response to comment #4: We based our study on a previously published US cost-effectiveness analysis in which a 20-year time horizon was used. In order to replicate the results of this previous study, it was necessary for us to also utilize a 20-year time horizon. In addition, the cost-effectiveness models used in the NICE and CADTH assessments of KEYNOTE-006 also used a partitioned survival modeling approach.

When modelling chronic conditions (such as cancer), or when treatments have differential effects on mortality, a lifetime horizon is most appropriate (NICE, CADTH, I.C.E.R.). Our survival extrapolations, based on published clinical data from the KEYNOTE-006 trial, indicate that approximately 10% of patients are expected to be alive at the 20-year timeframe. Therefore, using a shorter time horizon would result in important clinical events (e.g. disease progression, death) being missed, and the full costs and clinical benefits would not be captured. Our time horizon is therefore a reasonable approach, and a shorter time horizon would not be appropriate.

As per our response to comment #2 above, the following text was added to lines 363-371 in the Discussion section:

“When modelling chronic conditions (such as cancer), or when treatments have differential effects on mortality, a lifetime horizon is most appropriate. Our survival extrapolations, based on published clinical data from the KEYNOTE-006 trial, indicate that approximately 10% of patients are expected to be alive at the 20-year timeframe (Figure 3B). Using a shorter time horizon would result in important clinical events (e.g. disease progression, death) being missed, and the full costs and clinical benefits would not be captured. While a time horizon of 20 years may seem long, it is consistent with published literature[12] and health technology appraisal documents from NICE[9] and CADTH[10]. A shorter time horizon would therefore not be appropriate.”

  1. Reviewer’s comment #5: The model which the authors used is too simple, and it is difficult to evaluate the long-term changes of patients’ status.

Authors’ response to comment #5: We thank the reviewer for bringing up the topic of modeling methodology. We based our analysis on the methods reported in a previous cost-effectiveness analysis (Wang 2017) based on interim data from KEYNOTE-006. Our intent was to replicate this analysis and then reanalyze using longer-term follow-up data to assess the impact on the cost-effectiveness results. Since the previous cost-effectiveness publication reported a 3-state partitioned survival model, we also based our analysis on this approach in order to ensure comparability. The cost-effectiveness models used in the NICE and CADTH appraisals of KEYNOTE-006 also used a partitioned survival modeling approach.

Survival (OS, PFS) is the most important clinical endpoint in advanced cancer (Feigin 2004, Cheema 2013, Fiteni 2014, Delgado 2021), and partitioned survival modeling is a commonly used approach to model survival in oncology cost-effectiveness modeling (Woods 2020). While this approach may appear simple as the reviewer suggests, all of the published economic evaluation evidence for KEYNOTE-006 (Wang 2017, NICE assessment, CADTH assessment) have utilized the partitioned survival modeling approach. In addition, the Robert 2019 publication of long-term follow-up (5 years) from the KEYNOTE-006 trial provides clear documentation of the long-term changes of patients’ status, including both PFS and OS outcomes.

We agree that using a different and perhaps more complex modeling approach would be an interesting avenue to explore (we discussed this in lines 437-448 of the Discussion section. However, we found that our model accurately replicated the outcomes reported in KEYNOTE-006. Furthermore, using a more complex modeling approach would not necessarily improve model accuracy and could produce results that diverge from the from trial-reported outcomes (Gibson 2020). While such alternate modeling approaches would be useful additions to the totality of evidence, an investigation using a different modeling approach would necessarily be a different study with different objectives and is therefore outside the scope of our current study.

To address the reviewer’s comment on modeling methodology in the manuscript itself, we added a sentence in lines 377-380 of the Discussion section:

“In addition, although other approaches could be used to model outcomes from KEYNOTE-006, our model follows methodological recommendations from the NICE DSU 21 and is aligned with models from the UK (NICE HTA), Canada (CADTH HTA), and the US (Wang 2017)”.
